# An Insight into Deficient Mismatch Repair Colorectal Cancer Screening in a Romanian Population—A Bi-Institutional Pilot Study

**DOI:** 10.3390/medicina57080847

**Published:** 2021-08-20

**Authors:** Cristina Lungulescu, Vlad Mihai Croitoru, Simona Ruxandra Volovat, Irina Mihaela Cazacu, Adina Turcu-Stiolica, Dan Ionut Gheonea, Daniel Sur, Cristian Virgil Lungulescu

**Affiliations:** 1Doctoral School, University of Medicine and Pharmacy Craiova, 2 Petru Rares Str., 200349 Craiova, Romania; cristina.lungulescu@yahoo.com; 2Department of Oncology, Fundeni Clinical Institute, 258 Fundeni Str., 022238 Bucharest, Romania; vlad.m.croitoru@gmail.com (V.M.C.); irina.cazacu89@gmail.com (I.M.C.); 3Department of Medical Oncology, University of Medicine and Pharmacy Grigore T Popa Iasi, 700115 Iasi, Romania; simonavolovat@gmail.com; 4Department of Pharmacoeconomics, University of Medicine and Pharmacy of Craiova, 2 Petru Rares Str., 200349 Craiova, Romania; adina.turcu@gmail.com; 5Department of Gastroenterology, University of Medicine and Pharmacy Craiova, 2 Petru Rares Str., 200349 Craiova, Romania; digheonea@gmail.com; 6Department of Medical Oncology, The Oncology Institute “Prof. Dr. Ion Chiricuţă”, 400015 Cluj-Napoca, Romania; 711th Department of Medical Oncology, University of Medicine and Pharmacy “Iuliu Hatieganu”, 400012 Cluj-Napoca, Romania; 8Department of Oncology, University of Medicine and Pharmacy Craiova, 2 Petru Rares Str., 200349 Craiova, Romania; cristilungulescu@yahoo.com

**Keywords:** MSI, dMMR, colorectal, cancer, immunotherapy, Romania, incidence, PCR, IHC

## Abstract

*Background and Objectives*: Colorectal cancer (CRC) can be classified as mismatch-repair-deficient (dMMR) with high levels of microsatellite instability (MSI-H), or mismatch-repair-proficient (pMMR) and microsatellite stable (MSS). Approximately 15% of patients have microsatellite instability (MSI). MSI-H tumors are associated with a high mutation burden. Monoclonal antibodies that block immune checkpoints can induce long-term durable responses in some patients. Pembrolizumab is the first checkpoint inhibitor approved in the EU to treat dMMR–MSI-H metastatic CRC. *Materials and Methods*: Our study assesses the regional variability of MSI-H colorectal cancer tumors in Romania. Formalin-fixed, paraffin-embedded (FFPE) tissue blocks containing tumor samples from 90 patients diagnosed with colorectal cancer were collected from two tertiary referral Oncology Centers from Romania. Tissues were examined for the expression loss of MMR proteins (MLH1, PMS2, MSH2, MSH6) using immunohistochemistry or MSI status using polymerase chain reaction (PCR), respectively. *Results*: MSI-H was detected in 19 (21.1%) patients. MSI-H was located more in ascending colon (36.8% vs. 9.9%, *p*-value = 0.0039) and less in sigmoid (5.3% vs. 33.8%, *p*-value = 0.0136) than MSS patients. Most patients were stage II for MSI-H (42.1%) as well as for MSS (56.3%), with significant more G1 (40.9% vs. 15.8%, *p*-value = 0.0427) for MSS patients. Gender, N stage, and M stage were identified as significant prognostic factors in multivariate analysis. MSI status was not a statistically significant predictor neither in univariate analysis nor multivariate analysis. *Conclusion*: Considering the efficacy of PD-1 inhibitor in metastatic CRC with MSI-H or dMMR, and its recent approval in EU, it is increasingly important to understand the prevalence across tumor stage, histology, and demographics, since our study displayed higher regional MSI-H prevalence (21%) compared to the literature.

## 1. Introduction

As the third most common malignancy and the third leading cause of cancer-related mortality in both genders, colorectal cancer (CRC) poses a severe public health problem globally [1]. However, CRC can be one of the most curable diseases if it is discovered in early settings [2]. Colorectal cancer is a heterogeneous condition generated by the interaction of genetic and environmental components. Molecular variations that occur in CRC can be classified into three major categories: CIN (chromosomal instability), MSI (microsatellite instability), and CIMP (CpG Island Methylator Phenotype)—that causes gene function to be silenced by aberrant hypermethylation [3]. The purpose of this research is to examine MSI instability. Short (1–6 base pair) DNA repeating segments scattered across the entire genome are known as microsatellites or short tandem repeats (STR). Approximately 3 percent of the human genome is comprised of microsatellites, which are vulnerable to mutations due to their repeated structure [4]. An alternate-sized repeating DNA sequence that is not present in germline DNA is the hallmark of microsatellite instability in cancerous cells’ DNA. Microsatellite instability (MSI) represents a molecular phenotype caused by a defective DNA mismatch repair system (MMR). During DNA replication and recombination, mistakes such as base-base mismatches and insertions and deletions are corrected by the DNA mismatch repair mechanism. MMR proteins are fundamentally nuclear enzymes thatpromotethe repair of base-base mismatches that arise during cell proliferation by creating complexes (heterodimers) that adhere to aberrant DNA regions and initiate their removal [5]. MMR protein deficiency results in an accumulation of DNA replication defects, particularly in regions of the genome containing short repeating nucleotide sequences, which results in microsatellite instability. 

Approximately 15% of patients have microsatellite instability, according to twenty-two relevant publications with sample sizes ranging from 30 to 1000 and data on 7642 patients [5,6,7]. Three percent of the microsatellite instability-high (MSI-H) tumors have germline mutations in one of the MMR genes, defined as Lynch syndrome [8]. The remaining MSI-H tumors have acquired somatic mutations caused by abnormal methylation of the promoter of a gene that encodes a DNA MMR protein (MLH1).

Lynch syndrome (LS), alternatively referred to as hereditary non-polyposis colorectal cancer (HNPCC), is an inherited autosomal dominant condition that increases the risk of developing certain malignancies, particularly colorectal cancer. This is a consequence of a germline mutation in 1 of several genes involved in DNA mismatch repair (MMR), namely, MLH1, PMS2, MSH2, and MSH6 [9]. Since 90% of colorectal tumors due to LS have microsatellite instability, LS patients and their family members should undergo active surveillance;MSI testing could serve as a screening method.

Our study assesses the regional variability of MSI-H colorectal cancer tumors in Romania, as European Medicines Agency (EMA) recently approved immunotherapy as a treatment for metastatic colorectal cancer patients with high microsatellite instability (MSI-H) or mismatch repair deficiency (dMMR). Studying geographical variations and clinical characteristics of CRC patients is essential since innovative therapies, diagnosis techniques, and new methods of delivering treatments are constantly being developed [10,11,12].

## 2. Materials and Methods

Formalin-fixed, paraffin-embedded (FFPE) tissue blocks in which the tumor was visible macroscopically from 90 patients diagnosed with colorectal cancer were collected from two tertiary referral Oncology Centers from Romania. All patients included in this study are ethnic Romanians and of Caucasian descent. All patients received chemotherapy regimens combining fluoropyrimidines and oxaliplatin in an adjuvant setting. Following metastatic disease, targeted therapies such as Cetuximab/Panitumumab or Bevacizumab were added based on KRAS status. 

Thirty-three tissue samples were examined for the expression loss of MMR proteins (MLH1, PMS2, MSH2, MSH6) using immunohistochemistry (IHC). Positive staining was confirmed on adjacent normal tissue. MMR protein staining was deemed negative when all cancer cell nuclei failed to react with the antibody (dMMR). Samples missing one or more proteins were considered positive. If all 4 proteins were present, the likelihood of HNPCC/Lynch syndrome is reduced. 

Genomic DNA was extracted from the remaining 57 samples after macro-dissection from the cancer tissue, as follows: a certified gastrointestinal pathologist carefully evaluated and dissected the areas of the slides cut from FFPE tissue blocks representing the tumor and “normal” tissue—usually an uninvolved proximal or distal margin of resection. Analysis was carried out using five polymorphic markers (short tandem repeats—STR), referred to as the Bethesda panel, consists of two mononucleotide loci (Big Adenine Tract [*BAT*]-*25* and *BAT-26*) and three dinucleotide loci (*D2S123*, *D5S346*, and *D17S250*) [13]. Using this panel, tumors with instability at two or more of these loci were interpreted as MSI-high. In contrast, the lack of instability at either of the five loci was considered MSS.

### Statistical Analysis

Descriptive statistics were generated for all patients. The patients were divided into high microsatellite instability (MSI-H) and microsatellite stability (MSS). We compared the clinical characteristics of patients with MSI-H or MSS. Statistical comparations by microsatellite stability were assessed using Kruskal–Wallis (for continuous variables), Chi-Square (for categorical variables), or Log-rank (Mantel–Cox) test (for progression-free survival, PFS). The survival graph for PFS was generated using the Kaplan–Meier method. Time to event endpoints were analyzed using COX regression. The factors affecting survival (PFS) were identified using univariate and multivariate analysis. Statistical analysis was performed using GraphPad Prism 9.1.2 software (GraphPad Software, San Diego, CA, USA). The power analysis for our study was performed using G*Power 3.1.9.7. A two-sided *p*-value smaller than 0.05 was considered to be statistically significant.

## 3. Results

A total of 90 patients were enrolled. MSI-H was detected in 19 (21.1%) patients. Demographic and clinical characteristics of the patients in the MSI-H (*n* = 19) and MSS (*n* = 71) are summarized in Table 1. No significant statistical differences in age (*p* = 0.878) or gender (*p* = 0.514) were noted between the two groups. Moreover, no significant differences were found for TNM stages.

MSI-H was located more in ascending colon (36.8% vs. 9.9%, *p*-value = 0.0039) and less in sigmoid (5.3% vs. 33.8%, *p*-value = 0.0136) than MSS patients. Most patients were stage II for MSI-H (42.1%) as well as for MSS (56.3%), with significant more G1 (40.9% vs. 15.8%, *p*-value = 0.0427) for MSS patients, as Figure 1 shows.

No difference in PFS was noted between the two groups, as Table 2 and Figure 2 show.

Cox hazard regression was performed to identify the factors affecting the survival rate. In univariate analysis, males, advanced N stage, and M stage were statistically significant predictors of poor outcomes. MSI status didnot point out to be a statistically significant predictor neither in univariate analysis nor multivariate analysis. Gender, N stage, and M stage were identified as significant prognostic factors in multivariate analysis, as in Table 3.

We calculated the power for the outcomes from our study, and the obtained value of 90.8% demonstrates the sample size is representative ofour study’s results.

## 4. Discussion

Alteration in MMR proteins is frequently associated with the absence of an identifiable gene product, enabling IHC testing to indirectly determine the expression loss of the respective genes. IHC displays certain advantages over MSI analysis, such as relatively inexpensive and routinely used techniques. Moreover, it offers gene-specific information—the absence of a certain mismatch gene product (MLH1, MSH2, MSH6, or PMS2) can guide germline testing and aid in identifying patients with LS. However, IHC is susceptible to the quality of tissue preparation, variability of the antibodies, and interpretation—not a standardized method. Studies suggest that MSI testing and IHC are complimentary, as the loss of MMR protein expression is highly concordant with DNA-based MSI testing, providing a good sensitivity (>90%) and excellent specificity (100%) [14]. Several investigations revealed nearly perfect concordance between PCR and IHC tests. As a result, either approach is appropriate as a first-line screening method for determining dMMR/MSI-H status [15]. Additional criteria, such as more comprehensive family histories and genetic tests, including BRAF V600E mutation and hypermethylation of the MLH1 promoter, are necessary to differentiate between sporadic and hereditary colorectal cancer [16].

Our data showed noticeably higher regional MSI-H prevalence (21%) compared to other populations (10–13%) [4,5]. MSI-H was located more in ascending colon (36.8% vs. 9.9%, *p*-value = 0.0039) and less in the sigmoid (5.3% vs. 33.8%, *p*-value = 0.0136) than MSS patients, as reported in previous studies that have described that MSI-H is more frequently observed in proximal colon tumors than distal colon cancers.

The results of our study showed that gender, N stage, and M stage were identified as significant prognostic factors in multivariate analysis. These results support the notion that the TNM stage prevails as the gold standard for diagnosing colorectal tumors. However, numerous retrospective and population-based investigations have demonstrated that patients with dMMR tumors had a better stage-adjusted prognosis, implying that the superior outcomeassociated with dMMR CRCs is more apparent in early-stage lesions [7,17]. 

Patients with MSI-H CRC had a better prognosis, although it is unclear if MSI status predicts responsiveness to adjuvant chemotherapy [18]. Neither the univariate nor the multivariate analysis in the present study suggested that the MSI status significantly influenced prognosis.

The differentresponse of MSI-H tumors to chemotherapeutic drugs has been extensively studied in experiments. In addition to alkylating agents, DNA dMMR cells are resistant to platinum-containing treatments (cisplatin and carboplatin), antimetabolites (fluorouracil), and topoisomerase inhibitors (doxorubicin) [19]. According to findings, patients with stage II or stage III CRC with MSS tumors benefited from fluorouracil-based adjuvant treatment [20]. Cellular dynamics linked with MMR downregulation may explain these outcomes (increased apoptosis and decreased proliferation). However, multiple studies have been published that suggest MSI-H as a predictor of enhanced response to irinotecan or irinotecan-based chemotherapy in CRC patients.

Additionally, to quantify the response to chemotherapy, MSI has been recently established as a major predictive marker for immune checkpoint blockade response. Antitumor immune responses within MSI tumors are stronger than their MSS counterparts due to the high tumor mutational burden and neoantigen load that promote the infiltration of immune effector cells [21].

Nivolumab plus low-dose ipilimumab was authorized by the U.S. Food and Drug Administration on 11 July 2018, to treat MSI-H or dMMR metastatic colorectal cancer that has progressed after treatment with fluoropyrimidine, oxaliplatin, and irinotecan. The approval was based on findings from the Check-Mate-142 phase II investigation [22]. Subsequently, on 29 June 2020, the FDA approved pembrolizumab for the first-line treatment of patients with unresectable or metastatic colorectal cancer with high microsatellite instability or mismatch repair deficiency. The indication was approved based on the results of KEYNOTE 177 (NCT02563002), a trial in which 307 patients with previously untreated unresectable or metastatic MSI-H or dMMR colorectal cancer were included [23]. Similarly, on 10 December 2020, the European Medicines Agency’s (EMA’s) Committee for Medicinal Products for Human Use (CHMP) adopted the new indication for the medicinal product pembrolizumab. This marks the first approval of the CHMP for a target population defined by DNA repair deficiency biomarkers [24]. Pembrolizumab is recommended at a dose of 200 mg every three weeks or 400 mg every six weeks for MSI-H/dMMR colorectal cancer. Recently, on 20 May 2021, CHMP recommended nivolumab in combination with ipilimumab for the treatment of MSI-H/dMMR metastatic colorectal cancer patients who had previously received fluoropyrimidine-based combination therapy [25].

However, our findings show that MSI-H tumors had a lower rate of metastatic disease (10.5%) than MSS CRC (18.3%), highlighting the importance of future prospective large trials to demonstrate immunotherapy’s relevance in the adjuvant or neoadjuvant setting for CRC. FOLFOX6 adjuvant treatment with or without atezolizumab is currently being evaluated in phase III ATOMIC study (NCT02912559) to assess if the combined therapies offer a higher survival benefit than conventional chemotherapy alone for stage III dMMR CRC [26].

## 5. Conclusions

MSI is a surrogate marker of DNA mismatch repair deficiency and a surrogate for neoantigen load that enhances antitumor immune response. In light of evidence supporting the efficacy of PD-1 inhibitor in metastatic CRC with MSI-H or dMMR, and its recent approval in the EU, it is increasingly important to understand the prevalence across tumor stage, histology, and demographics, since our study displayed higher regional MSI-H prevalence (21%) compared to the literature.

## Figures and Tables

**Figure 1 medicina-57-00847-f001:**
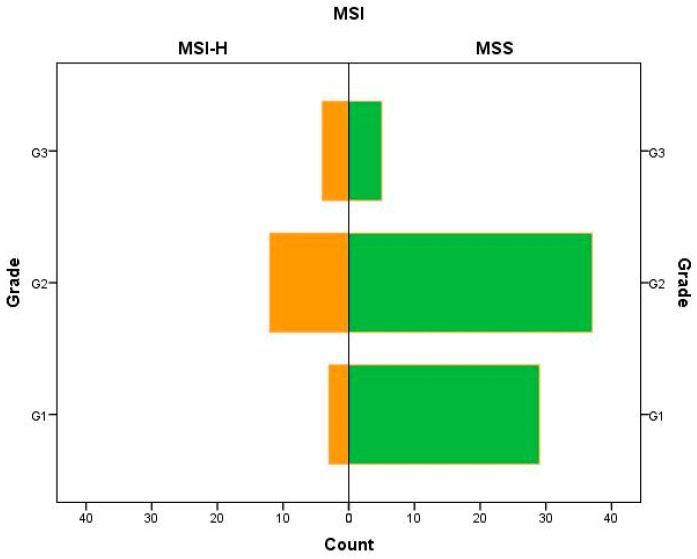
Patients’ distribution for histologic grade by MSI.

**Figure 2 medicina-57-00847-f002:**
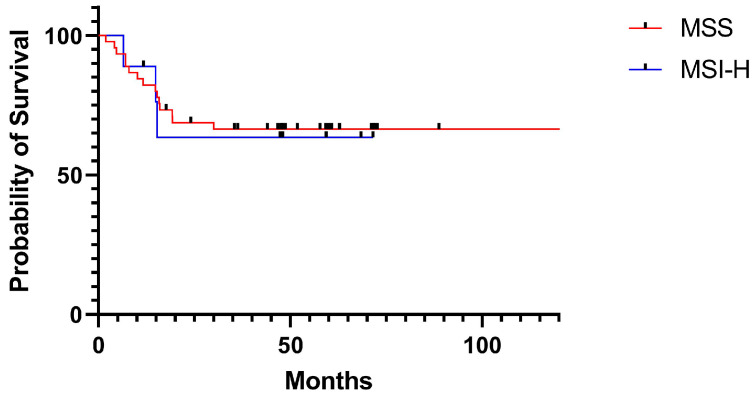
Kaplan– survival curves for patients with colorectal carcinoma by MSI status. MSI, microsatellite instability; MSI-H, high microsatellite instability; MSS, microsatellite stability.

**Table 1 medicina-57-00847-t001:** Characteristics of the cohort used in this study.

Variable	Patients			*p*-Value
	Total (*n* = 90)	MSI-H (*n* = 19)	MSS (*n* = 71)	
Age				0.878 ^1^
Mean (SD)	61.8 (10.0)	61.8 (10.7)	61.8 (9.9)	
Median (IQR)	62 (54–67)	62 (52–67)	63 (58–67)	
Range	36–84	45–84	36–83	
Gender, female, *n* (%)	40 (44.4%)	8 (42.1%)	32 (45.1%)	0.99 ^2^
Tumor location, *n* (%)				-
Ascending colon	14 (15.6%)	7 (36.8%)	7 (9.9%)	
Cecum	8 (8.9%)	1 (5.3%)	7 (9.9%)	
Descending colon	12 (13.3%)	3 (15.8%)	9 (12.7%)	
Hepatic angle	1 (1.1%)	0	1 (2.2%)	
Recto-sigmoid	8 (8.9%)	1 (5.3%)	7 (9.9%)	
Rectum	13 (14.4%)	2 (10.5%)	11 (15.5%)	
Sigmoid	25 (27.8%)	1 (5.3%)	24 (33.8%)	
Superior rectum	3 (3.3%)	1 (5.3%)	2 (2.8%)	
Transverse colon	6 (6.7%)	3 (15.8%)	3 (4.2%)	
Tumor location, *n* (%)				0.0025 ^2,^*
Proximal	33 (36.7%)	13 (68.4%)	20 (28.2%)	
Distal	57 (63.3%)	6 (31.6%)	51 (71.8%)	
Disease stage				0.447 ^2^ (I-II vs III-IV)
I	2 (3.6%)	1 (5.3%)	1 (1.4%)	
II	48 (63.6%)	8 (42.1%)	40 (56.3%)	
III	21 (16.4%)	7 (36.8%)	14 (19.7%)	
IV	19 (16.4%)	3 (15.8%)	16 (22.5%)	
Histologic Grade, *n* (%)				0.052 ^2^
G1	32 (35.6%)	3 (15.8%)	29 (40.9%)	
G2	49 (54.4%)	12 (63.2%)	37 (52.1%)	
G3	9 (10.0%)	4 (21.1%)	5 (7%)	
T-Stage, *n* (%)				0.99 ^2^ (T1–2 vs. T3–4)
T1	1 (1.1%)	0	1 (1.4%)	
T2	6 (6.7%)	1 (5.3%)	5 (7.0%)	
T3	63 (70.0%)	14 (73.7%)	49 (69.0%)	
T4	13 (14.4%)	4 (21.1%)	9 (12.7%)	
Tx	7 (7.8%)	0	7 (9.9%)	
N-Stage, *n* (%)				0.796 ^2^ (N0 vs N1–2)
N0	51 (56.7%)	10 (52.6%)	41 (57.7%)	
N1	18 (20.0%)	4 (21.1%)	14 (19.7%)	
N2	12 (13.3%)	5 (26.3%)	7 (9.9%)	
Nx	9 (10.0%)	0	9 (12.7%)	
M-Stage, *n* (%)				0.223 ^2^ (M0 vs M1)
M0	70 (77.8%)	17 (89.5%)	53 (74.6%)	
M1	3 (3.3%)	0	3 (4.2%)	
M1 with hepatic metastases	11 (12.2%)	1 (5.3%)	10 (14.1%)	
M1 with hepatic, pulmonary metastases	3 (3.3%)	1 (5.3%)	2 (2.8%)	
M1 with pulmonary metastases	1 (1.1%)	0	1 (1.4%)	
Mx	2 (2.2%)	0	2 (2.8%)	
Metastatic CRC, yes, *n* (%)	15 (16.7%)	2 (10.5%)	13 (18.3%)	0.729 ^2^

All percentages are based on the total number of patients in each group. CRC, colorectal cancer. ^1^ Kruskal–Wallis *p*-value; ^2^ Fisher’s exact test. *p*-value; *, significant difference.

**Table 2 medicina-57-00847-t002:** PFS comparison between MSI-H and MSS patients.

PFS	Patients			*p*-Value
	Total (*n* = 55)	MSI-H (*n* = 9)	MSS (*n* = 46)	
Mean (SD)	42.2 (27)	38.1 (26.04)	42.9 (27.4)	0.865 ^1^
Median (IQR)	47.5 (15.3–60.1)	47.3 (13.3–63.9)	47.6 (15.9–60.2)	
Range	1.9–128.8	6.48–71.5	1.9–128.8	

^1^ Log-rank (Mantel–Cox) test *p*-value; PFS, progression-free survival; SD, standard deviation; IQR, interquartile range.

**Table 3 medicina-57-00847-t003:** Cox proportional hazard regression for clinical characteristics.

Factors		Univariate Analysis		Multivariate Analysis	
		Hazard Ratio	*p*-Value	Hazard Ratio	*p*-Value
Age		1.05 (0.98–1.12)	0.21	-	-
Gender	Male vs. female	30.63 (2.61–359.04)	0.006	5.33 (1.24–22.98)	0.025
MSI	MSS vs. MSI-H	8.24 (0.92–73.54)	0.059	3.14 (0.7–14.16)	0.136
Location	Proximal vs. distal	3.76 (0.55–25.53)	0.175	-	-
Stage	III-IV vs. I-II	0.09 (0.01–1.59)	0.1	-	-
T stage	T3–4 vs. T1–2	2.47 (0.32–19.2)	0.388	-	-
N stage	N0 vs. N1–2	0.01 (0.0–0.53)	0.023	0.01 (0.001–0.104)	<0.001
M stage	M0 vs. M1	0.25 (0.07–0.91)	0.036	0.19 (0.05–0.7)	0.012

## Data Availability

The datasets used and/or analyzed during the current study are available from the corresponding author on reasonable request.

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
