# Peer review of "An Insight into Deficient Mismatch Repair Colorectal Cancer Screening in a Romanian Population—A Bi-Institutional Pilot Study"

_medicina, 2021, doi:10.3390/medicina57080847_

Round 1

Reviewer 1 Report

Dear Authors,

Your manuscript focused on an important topic of the targeted treatment of colon cancer, and it takes into  acknowledged recent findings directions.

Knowing the percentage of occurrence of MSI in given population is crucial, and needs to be taken into account while planning and developing highly specialised treatment centres.

The structure of your manuscript does meet with current rules.

I would like to congratulate the clear presentation of data (Table 1), this form of presentation certainly helps understanding the relations. I do hope that this valuable research will be continued. The reason for higher occurrence of MSI-H is especially interesting (line 167). It would be worth to look at data on survival.

The manuscript raises a problem of CRC, which is the third most prevalent malignancy and the third most frequent cause of cancer related mortality in both sexes. The aim of the study is the regional variability of MSI-H colorectal cancer tumors in Romania.

It was a pleasure to review this manuscript. The accurate characterisation of groups of patients, clearly presented results, brief comparison of IHC and MSI methods and exhaustive discussion are strengths of this review.

The manuscript would benefit from including data on patient’s ethnicity and socioeconomic background, which is known to be a factor in occurrence of various malignancies. Survival rate among patients from wealthier backgrounds could be investigated in further studies.
It might be worth clarifying an abbreviation in line 75.

Apart from this detail, this manuscript is well written.

I look forward to seeing your further research.

Author Response

Please see the attachment. Thank you for your time reviewing our manuscript.

Reviewer 2 Report

In this manuscript, the authors provide a simple description of 90 colorectal cancer patients from Romania. The goal of the study as stated in the introduction was to assess "the regional variability of MSI-H colorectal cancer tumors in Romania", which I'm assuming is referring to geographical locations given only relevant line in the discussion [line 167]. Our understanding of genomics of CRC related to MMR/MSI status, heterogeneity of MMR, response to immunotherapy, is already more advanced. There are datasets of 100s/1000s published correlating much more clinical factors than what is described. Thus, the study is too simple and is of a very small cohort, and does not introduce any new findings or note anything special about this cohort in particular. The only new data is that there may be 'slightly higher' prevalence in this cohort. but this is not definite due to the small sample size.

some general notes:

methods: for such small numbers, please use Fisher test instead of chi-squared for calculating p-value. some of the sub-groups have too few samples (n=1??) and likely make the statistical analysis more noisy

was any assessment of sporadic vs germline were any of the deficient cases due to Lynch?

paragraph 1: doesn't flow well. jumps through many topics without transitions.

what was the purpose of discussion IHC vs MSI-PCR? In this study it seemed that half samples used IHC and the other have PCR, and no comparison can be made of their concordance.

what were the other populations that had 10-13% MSI-H prevalence? Given the small cohort in this case, is the regional difference significantly higher or just not enough samples to be representative? Are there any cohort or genomic or clinical differences that could explain this difference? The prevalence in other populations has been reported to be higher for earlier stages of colorectal cancer, and up to 20% which is similar to the prevalence here.

line 172-183: did all patients get the same adjuvant chemo? the response of MSI tumors to chemo seen in other studies may be due to the different cell types (the reference is very old, and temozolomide resistance in MMR-D tumors is now well described in glioma patients, for example).

Reviewer 3 Report

The authors present a good study to report the MSI/MSS status in Romanian patients with colorectal cancer.
A few points may be considered:
As the authors are trying to link the MSI and the clinical use of Pembrolizumab, it may be better to mention the therapy given to the patients they have presented. That should be important in the survival analysis. Proportion of metastatic CRC was lower in MSI although sample size was very small to comment. However, non-metastatic CRC could have analyzed separately than metastatic CRC to see the prognostic significance of MSI. This has been reported to be different in non-metastatic and metastatic setting and also influenced by therapy used.
The multivariate HR shown in the table-3 may be a little confusing without explaining the reference category clearly. For example, if HR of 5.33 for gender (male vs female) indicates poor outcome in male, then why the multiple HR of 0.01 for N-stage (N1-2 vs N0) is interpreted as poor outcome in N1-2? The authors either coded the variable in different order or this is the effect of different therapeutic regimen. Same issue with M-stage.
I guess, if the authors want to link the recent evidence of effectiveness of Pembrolizumab in metastatic CRC (as they have mentioned in introduction and conclusion) and their present study finding of prevalence of MSI among CRC patients in Romania,  they also should take into account the fact that lower proportion of MSI tumors were metastatic in nature. Therefore, only a small proportion of CRC patients are suitable for such therapy. Trial of Pembrolizumab in non-metastatic MSI setting will be required to comment on significance of MSI detection in clinical therapeutic use.

Minor edit in sentence: Line 30: “and” may be replaced by “or”. Line 181-182: May consider changing to “Neither the univariate, nor the multivariate analysis in present study suggested that the MSI status had significant influence on prognosis”
